**Data Availability Statement:** All relevant data are within the paper and its Supporting information files.

# Direct cost of systemic arterial hypertension and its complications in the circulatory system from the perspective of the Brazilian public health system in 2019

**Daniel da Silva Pereira Curado**[1]*, **Dalila Fernandes Gomes**[1], **Thales Brendon Castano Silva**[2], **Paulo Henrique Ribeiro Fernandes Almeida**[3], **Noemia Urruth Leão Tavares**[1], **Camila Alves Areda**[4], **Everton Nunes da Silva**[1,4]

1 Graduate Program in Public Health, University of Brasilia, Brasilia, Federal District, Brazil, 2 Faculty of Pharmacy, Department of Medicine, Federal University of Bahia, Salvador, Bahia, Brazil, 3 Faculty of Pharmacy, Graduate Program in Medicines and Pharmaceutical Services, Federal University of Minas Gerais, Belo Horizonte, Minas Gerais, Brazil, 4 Faculty of Ceilandia, University of Brasilia, Brasilia, Federal District, Brazil

* danielspcurado@gmail.com

## Abstract

### Introduction

Systemic arterial hypertension (SAH), a global public health problem and the primary risk factor for cardiovascular diseases, has a significant financial impact on health systems. In Brazil, the prevalence of SAH is 23.7%, which caused 203,000 deaths and 3.9 million DALYs in 2015.

### Objective

To estimate the cost of SAH and circulatory system diseases attributable to SAH from the perspective of the Brazilian public health system in 2019.

### Methods

A prevalence-based cost-of-illness was conducted using a top-down approach. The population attributable risk (PAR) was used to estimate the proportion of circulatory system diseases attributable to SAH. The direct medical costs were obtained from official Ministry of Health of Brazil records and literature parameters, including the three levels of care (primary, secondary, and tertiary). Deterministic univariate analyses were also conducted.

### Results

The total cost of SAH and the proportion of circulatory system diseases attributable to SAH was Int\$ 581,135,374.73, varying between Int\$ 501,553,022.21 and Int\$ 776,183,338.06. In terms only of SAH costs at all healthcare levels (Int\$ 493,776,445.89), 97.3% were incurred in primary care, especially for antihypertensive drugs provided free of charge by the Brazilian public health system (Int\$ 363,888,540.14). Stroke accounted for the highest cost

**Funding:** The authors received no specific funding for this work.

**Competing interests:** The authors have declared that no competing interests exist.

attributable to SAH and the third highest PAR, representing 47% of the total cost of circulatory diseases attributable to SAH. Prevalence was the parameter that most affected sensitivity analyses, accounting for 36% of all the cost variation.

## Conclusion

Our results show that the main Brazilian strategy to combat SAH was implemented in primary care, namely access to free antihypertensive drugs and multiprofessional teams, acting jointly to promote care and prevent and control SAH.

## Introduction

Systemic arterial hypertension (SAH) also known as high blood pressure ($\geq$140/90 mmHg) [1–3] is a public health problem in low- and middle-income countries (LMICs) [4,5]. It is considered the main risk factor for cardiovascular diseases [6–9]. According to the World Health Organization (WHO), if left untreated, SAH can cause dementia, kidney failure and blindness [10].

In 2015, the prevalence of SAH was 24.1% in men and 20.1% in women, affecting around 1.13 billion people worldwide [11]. In that same year, an overall prevalence of SAH of 32.3% was found in LMICs [12]. In Brazil, studies conducted between 2013 and 2019 indicate that the prevalence of SAH in adults ranges between 21.4% and 32.3%, depending on the methodology used to identify and measure blood pressure [13–15]. Due to the increasing prevalence of high blood pressure, it will be difficult for LMICs to reach the world goal of a 25% reduction by 2025 [11], as recommended by the WHO [16]. Estimates suggest that high blood pressure caused around 7.8 million deaths and 143 million disability-adjusted life years (DALYs) worldwide in 2015, since it is a risk factor primarily for ischemic heart disease and hemorrhagic and ischemic stroke. In Brazil, there were an estimated 203,000 deaths and 3.9 million DALYs in 2015 [9]. Although the control of SAH has improved in high-income countries [17,18], it has declined slightly in LMICs [19].

In addition to human suffering, SAH has important financial consequences for health systems [10,20,21]. In the USA, where the prevalence of SAH is 36.9%, it is estimated that in 2018 the annual medical costs of hypertensive patients were US$ 131 billion higher than those of non-hypertensive patients [22]. In 2014, the costs attributable to SAH for the British health system were estimated at £ 2.1 billion, considering an SAH prevalence of 30% [23]. In Canada, with an SAH prevalence of 23%, the estimated direct medical cost for the public health system in 2020 was C$ 20.5 billion attributable to SAH [24].

The estimated annual direct cost of SAH for the Brazilian public and private health systems was US$ 671.6 million, which represented 0.08% of the gross domestic product (GDP) in 2005 and 1.1% of total health costs [25]. Another Brazilian study estimated only the direct costs for public health, finding a cost attributable to SAH of BRL 2.03 billion (US$ 523.7 million) in 2018 [26]. Neither study conducted a comprehensive analysis of primary care costs in the public health system, including the free provision of antihypertensive drugs, and consultations with physicians and non-physician health workers (NPHWs) at health units and home visits.

The aim of this study was to estimate the cost of SAH and SAH-related circulatory system diseases to the Brazilian public health system (SUS acronym in Portuguese) in 2019, including the costs of primary, secondary, and tertiary healthcare.

## Methods

### Study setting

The 1988 Federal Constitution established health as a universal right and the obligation of the state. This led to the creation of the Unified Health System (SUS), adhering to the principles of universality, integrality and community participation [27]. The SUS is publicly funded by the federal, state, and municipal governments. The health services are free of charge [28]. SUS users have access to a wide range of health services, including primary, secondary, and tertiary care. In primary care, patients have access to Family Health Strategy (FHS) teams, consisting of physicians, nurses, nurse technicians and community health agents [29]. These teams promote the prevention and control of chronic (including SAH) and infectious diseases, as well as health surveillance. In addition, the SUS provides essential medication based on a national drug catalog (including antihypertensives, such as hydrochlorothiazide, losartan, captopril, enalapril, atenolol, amlodipine, propranolol, furosemide and nifedipine) [30] free of charge at public and private pharmacies accredited in the Brazilian Popular Pharmacy Program (PFPB) [31]. The PFPB was created to broaden access to essential drugs, subsidized by the Ministry of Health of Brazil. Thus, essential drugs for SAH, diabetes and asthma are provided at no cost [32–34]. Although there is a legal obligation to provide drugs free of charge, the population has faced barriers to access to medication in Brazil, not achieving full coverage [35]. In addition, Family Health Support Centers (NASF) act in an integrated manner with NPHWs in primary care. This allows joint assessment of cases, shared care among professionals and the creation of therapeutic plans that optimize locally-adopted interventions. Staff at these centers include mental health, rehabilitation, nutrition, maternal and child care, pharmacy and social assistance professionals [36]. In secondary specialized care, examinations, consultations with specialists and hospitalizations for low-severity cases are provided. This specialized care is generally offered in medical clinics (Emergency Care Units–UPAs). Tertiary care involves highly complex and technologically advanced procedures provided in a hospital setting. Currently, 75% of Brazilians depend exclusively on the SUS for healthcare [37].

### Study design

A prevalence-based cost-of-illness was conducted to estimate the direct medical-hospital costs of SAH and circulatory system diseases for which this condition is considered a risk factor, using the Population Attributable Risk (PAR). The top-down approach was used, based on Ministry of Health of Brazil records, to identify, measure and quantify primary, secondary, and tertiary care costs. This study followed national and international recommendations for cost-of-illness studies [38,39].

Costs were estimated based on the prevalence of SAH in Brazil, including all the people treated in the SUS, irrespective of the severity and time since disease onset [15]. Moreover, only the adult population was considered (≥20 years of age) in analysis [40], which was obtained from the main national survey [15]. This study was conducted from the perspective of the SUS and costs were adjusted to 2019 prices. Thus, non-medical direct costs (patient transport, caregiver payments), indirect costs (absenteeism, presenteeism and premature death) and intangible costs (pain/suffering) were not considered. The costs were collected in BRL and later adjusted for purchasing-power parity (PPP) in 2018 (Int$ 1 = BRL 2.20), the latest year available from the World Bank [41].

## SAH costs

The costs of SAH includes the services provided by the three healthcare levels (primary, secondary, and tertiary). Primary care was included because the main SUS strategies for prevention, diagnosis and control of SAH occur at this level [42].

The costs of SAH in secondary and tertiary care were obtained from the Outpatient Information System (SIA/SUS) and Hospital Information System (SIH/SUS). Both systems are national and provide the values reimbursed by the Ministry of Health of Brazil to the health services that performed the medium- and high-complexity procedures, including medical consultations, NPHW care, drugs administered at health units, hospitalizations, surgeries, support care, complementary procedures and laboratory and imaging exams. The data were extracted using the 10th version of the International Classification of Diseases (ICD-10) in terms of SAH (I10). The data were analyzed using TabWin software, created by the Ministry of Health of Brazil [43].

The costs of SAH in primary care include expenses on medical and NPHW consultations and antihypertensive drugs. The number of medical and NPHW consultations for SAH was obtained from the Health Information System for Primary Care (SISAB/SUS) [44]. A value of Int$ 4.54 was used for medical consultations and Int$ 2.86 for NPHW consultations; Int$ 1.43 was added for home visits [45].

There is no public national registry of the number of drugs dispensed to hypertensive patients in primary care. Thus, our estimate was based on epidemiological data and public access to the National Survey on Access, Use and Promotion of Rational Use of Medicines (PNAUM) [15,46]. For each drug, the daily defined dose (DDD) [47] was multiplied by 365 days in order to obtain annual intake. The cost per pharmaceutical unit was obtained from i) the Health Prices Database (BPS) [48] for drugs dispensed in public pharmacies (hydrochlorothiazide, losartan, captopril, enalapril, atenolol, amlodipine, propranolol, furosemide and nifedipine), accessed on 07/24/2020, and ii) the reference price for drugs dispensed in private pharmacies of the PFPB (hydrochlorothiazide, losartan, captopril, enalapril, atenolol and propranolol) [49]. The BPS is a mandatory registry system that allows public access to government purchases of drugs and other health-related products in Brazil. The weighted average of the price of each drug was obtained from this database, the weight factor being the number of drugs acquired up to 07/24/2020. Since the drugs amlodipine, furosemide, and nifedipine are not available in PFPB, the prices charged at public pharmacies were used. The number of hypertensive individuals under drug treatment was obtained from the literature [15]. The distribution of antihypertensive drugs in use by the population in the SUS was calculated based on the primary data of the PNAUM [46]. We considered the cases in which patients used up to three drugs concomitantly. Thus, the distribution was as follows: 54.7% of hypertensive patients under drug treatment used only one; 37.4% two and 7.9% three antihypertensive drugs. Based on the same study [46], it was also possible to identify the distribution of each drug used by hypertensive patients, stratified by the number (1, 2 or 3 drugs). The complete description of the parameters used in the analysis can be found in S1–S4 Tables.

## Costs of SAH-related circulatory system diseases

To calculate the cost of SAH-related circulatory system diseases, the PAR was applied to the costs obtained after consulting the SIA/SUS and SIH/SUS. The PAR was calculated using the

following formula [38,50]:

$$PAR = \frac{P(RR - 1)}{P(RR - 1) + 1}$$

P = Prevalence of people with SAH in Brazil

RR = Measure of effect size (Relative Risk, Odds Ratio or Hazard Ratio) of individuals with SAH who developed circulatory system diseases versus those without SAH

The prevalence of SAH in Brazil (23.7%) was obtained from the PNAUM [15]. The PNAUM was a national population-based cross-sectional study that assessed the use of drugs, including by hypertensive patients [51].

The literature reports that SAH is a risk factor for circulatory system diseases [9]. In order to obtain the measures of effect size (Relative Risk, Odds Ratio or Hazard Ratio) of circulatory system diseases, an overview of systematic reviews with meta-analysis indexed in international electronic databases was conducted (Cochrane Database of Systematic Reviews, Embase and PubMed), with no restriction for language or year of publication. The following eligibility criteria were adopted: systematic reviews or pooled analysis studies that related SAH with the selected diseases, and that provided measures of effect size. The eligible articles were screened by two pairs of researchers (DSPC and PHRFA; TBCS and DFG). The article selection process, including the search strategy used, can be found in S5 Table and S1 Fig. After the potential eligible articles were identified, two independent researchers (DSPC and TBCS) assessed the quality of the articles selected using the Assessment of Multiple Systematic Reviews 2 (AMSTAR 2) [52] (S6 Table).

Based on the overview performed, we included the following diseases: coronary artery disease (I25.1), heart failure (I50), subarachnoid hemorrhage (I60), intracerebral hemorrhage (I61), stroke (I64), carotid atherosclerosis (I65.2 and I70.8), abdominal aortic aneurysm (I71.3 and I71.4), and peripheral artery disease (I73.9).

The SIA/SUS and SIH/SUS were used to obtain direct costs between January and December 2019, in relation to the ICD-10 codes of circulatory system diseases attributable to SAH. These costs were multiplied by the PAR, resulting in the costs of SAH-related circulatory system diseases.

### Sensitivity analysis of the estimated costs

Deterministic univariate sensitivity analyses were carried out to assess the uncertainties of the parameters that could affect the cost estimates of SAH and circulatory system diseases attributable to the condition in the SUS in 2019 [53]. The selection of the parameters assessed was based on the literature (prevalence and measures of effect size of diseases associated with SAH) and the presence of a variation in the data on the use of drugs in primary care (distribution of the use of antihypertensives and their respective costs).

For the prevalence of SAH in Brazil, the upper and lower limits (32.3 and 21.4%, respectively) of another Brazilian study [13] were used in order to explore the discrepancies in the literature regarding this prevalence in Brazil. The variation in prevalence affects the population using antihypertensives in primary care, the number of medical consultations and treatments by NPHWs of hypertensives in primary care and the PAR of all diseases associated with SAH. The variation in the distribution of antihypertensives in primary care was based on the 95% confidence intervals of the PNAUM data analyzed (S4 Table) and influences the total costs of antihypertensives in primary care. The costs of public pharmacy drugs were varied using the 1st to 3rd quartiles of the BPS [48] for each of the antihypertensive drugs, in order to avoid

discrepant values. In the case of the PFPB, the minimum and maximum reference values reimbursed to the Brazilian states were used [49]. The cost variations of both supply sources (public pharmacies and PFPB) affect only the costs of antihypertensives in primary care. The measures of effect size were varied based on their 95% confidence intervals reported in the same articles used for the case base, affecting the PAR and consequently, the costs of circulatory system diseases attributable to SAH. The results of deterministic univariate sensitivity analyses were summarized and presented in a tornado diagram.

### Ethical aspects

Given that this study used only secondary data available in public databases, approval from the Research Ethics Committee of the National Research Ethics Commission (CEP/CONEP) was not required [54].

## Results

### SAH costs

Fig 1 presents the estimated number of hypertensive Brazilian adults ≥20 years of age in 2019, stratified by type of drug supply, the number of concomitant antihypertensives and type of drug in use. The hypertensive Brazilian population using antihypertensive drugs was estimated at 28,118,310 individuals in 2019. Of these, an estimated 74.3% obtained antihypertensives free of charge from the public health system. The populations calculated for each stratum are exhibited in S1 and S4 Tables.

The cost of antihypertensives at public pharmacies was around Int$ 169.7 million, and approximately Int$ 194.2 million in the PFPB in 2019. With respect to clinical services, 70.9% (n = 19,864,881) of the consultations/care of hypertensive patients in primary care were conducted by physicians in Brazil in 2019. These medical consultations represented a cost of approximately Int$ 91.5 million. In the case of NPHW care, the cost in 2019 was around Int$ 24.9 million. In outpatient and hospital treatments in secondary and tertiary care, the costs of SAH were approximately Int$ 4.7 million and Int$ 8.8 million, respectively (Table 1).

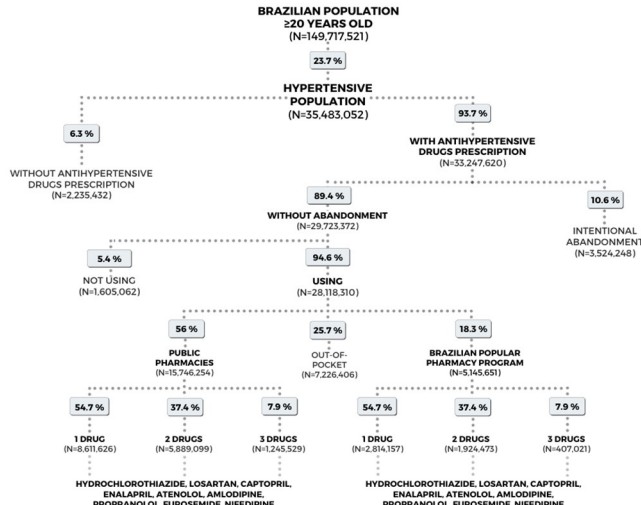

**Fig 1. Estimated population using antihypertensives provided by the Brazilian public health system in 2019.**

**Table 1. Estimated direct costs of SAH in the SUS.** Brazil, 2019.

| | Number of individuals/procedures (≥20 years of age) | Costs (Int$) |
|---|---|---|
| Primary care | | |
| Public pharmacy drugs | 15,746,254 individuals | 169,689,247.42 |
| Brazilian Popular Pharmacy Program drugs | 5,145,651 individuals | 194,199,292.72 |
| Medical consultation | | |
| At health units | 19,035,101 consultations | 86,523,186.36 |
| At home | 829,780 consultations | 4,956,050.64 |
| Non-physician health worker care | | |
| At health units | 7,055,235 consultations | 20,203,627.50 |
| At home | 1,096,538 consultations | 4,705,144.87 |
| Total primary health costs | | 480,276,548.52 |
| Total outpatient costs of SAH in secondary and tertiary care | 443,817 procedures | 4,657,011.35 |
| Total hospital costs of SAH in secondary and tertiary care | 53,024 procedures | 8,842,886.02 |
| Total costs of SAH in the SUS | | 493,776,445.89 |

Source: [48] Health Prices Database (BPS); [15] National Survey on Access, Use and Promotion of Rational Use of Medicines (PNAUM); [49] Ordinance No. 739, of March 27, 2018, which updates the reference values of drugs provided by the Brazilian Popular Pharmacy Program Here to treat hypertension, diabetes mellitus and asthma; [45] System for Managing the Table of Procedures, Drugs and Orthoses, Prostheses and Special Materials of the SUS (SIGTAP); [44] Health Information System for Primary Care (SISAB/SUS); [43] Outpatient Information System (SIA/SUS) and Hospital Information System (SIH/SUS).

## Costs of SAH-related circulatory system diseases

Table 2 presents the measures of effect size of the association between SAH and circulatory system diseases identified in the overview conducted in the present study, the corresponding ICD-10 codes and the PAR calculated for each of the diseases. The complication that had the greatest measure of effect size related to SAH and consequent highest PAR value was

**Table 2. Measures of effect size, corresponding ICD-10 code and Population Attributable Risk of circulatory system diseases associated with SAH.**

| Associated disease | ICD-10 | | Measure of effect size | | Population Attributable Risk | | Ref. |
|---|---|---|---|---|---|---|---|
| | | | Base case | 95%CI | Base case | 95%CI | |
| Coronary artery disease | I25.1 | OR | 1.61 | 1.37 to 1.89 | 0.13 | 7.34 to 22.33 | [55] |
| Heart failure | I50 | HR | 1.61 | 1.33 to 1.96 | 0.13 | 6.60 to 23.67 | [56] |
| Subarachnoid hemorrhage | I60 | OR | 2.60 | 2.00 to 3.10 | 0.27 | 17.63 to 40.42 | [57] |
| Intracerebral hemorrhage | I61 | OR | 3.77 | 2.58 to 5.51 | 0.40 | 25.27 to 59.30 | [58] |
| Stroke | I64 | OR | 3.50 | 3.18 to 3.85 | 0.37 | 31.81 to 47.93 | [59] |
| Carotid atherosclerosis | I65.2, I70.8 | OR | 1.81 | 1.55 to 2.13 | 0.16 | 10.53 to 26.74 | [60] |
| Abdominal aortic aneurysm | I71.3, I71.4 | RR | 1.66 | 1.49 to 1.85 | 0.14 | 9.49 to 21.54 | [61] |
| Peripheral artery disease | I73.9 | OR | 1.67 | 1.50 to 1.86 | 0.14 | 9.67 to 21.74 | [62] |

Note: ICD-10: International Classification of Diseases; 95%CI: 95% confidence interval; OR: Odds Ratio; HR: Hazard Ratio; RR: Relative Risk. The prevalence values used in the PAR were 23.7% in the case base [15], 21.4% in the lowest 95%CI and 32.3% in the highest 95%CI [13]. Source: (55) Poorzand et al, 2019; [56] Yang et al, 2015; [57] Feigin et al, 2005; [58] Ariesen et al, 2003; [59] Wang et al, 2017; [60] Ji et al, 2019; [61] Kobeissi et al, 2019; [62] Song et al, 2019.

intracerebral hemorrhage, followed by stroke and subarachnoid hemorrhage. The other complications of SAH exhibited similar PAR values (Table 2).

Table 3 presents the costs of SAH-related circulatory system diseases, stratified by the type of care (outpatient or hospital). The costs of these diseases are concentrated in hospital care, where highly complex treatment is commonly provided. In addition, the associated disease that incurred the highest costs for the SUS was stroke, in both outpatient and hospital care. Finally, the total estimated cost of diseases attributable to SAH in Brazil, considering both types of care, was approximately Int$ 87.4 million. This value represents 15% of the total cost of SAH and circulatory system diseases attributable to the condition in 2019, of Int$ 581,135,374.73.

Intracerebral hemorrhage was the disease with the highest PAR related to SAH (Table 2). However, the disease associated with the highest cost attributable to SAH was stroke, with the second highest PAR value (Tables 2 and 3). Stroke accounted for 47% of the total cost of SAH-related circulatory system diseases. In addition, despite exhibiting the lowest PAR, heart failure incurred the second highest cost attributable to SAH (Tables 2 and 3).

## Sensitivity analysis of estimated costs

The tornado diagram (Fig 2) presents the decreasing order of the variation in parameters that affected the total cost of SAH and circulatory system diseases attributable to this condition. A large part of the variation in total costs is concentrated on the right side of the graph, that is, above the base case value of Int$ 581 million. The values varied between Int$ 501.5 million, the

**Table 3. Cost of SAH-related circulatory system diseases in secondary and tertiary care in the SUS.** Brazil, 2019.

| | ICD-10 | Number of procedures | Total cost of each disease (Int$) | Cost attributable to SAH (Int$) |
|---|---|---|---|---|
| **Outpatient care** | | | | |
| Coronary artery disease | I25.1 | 20,109 | 4,174,254.49 | 527,247.76 |
| Heart failure | I50 | 4,997 | 3,202,289.70 | 404,479.43 |
| Subarachnoid hemorrhage | I60 | 282 | 222,730.13 | 61,237.87 |
| Intracerebral hemorrhage | I61 | 139 | 103,014.18 | 40,825.95 |
| Stroke | I64 | 658,431 | 5,536,438.76 | 2,059,868.11 |
| Carotid atherosclerosis | I65.2, I70.8 | 8,114 | 313,434.96 | 50,479.55 |
| Abdominal aortic aneurysm | I71.3, I71.4 | 5,586 | 191,286.17 | 25,873.80 |
| Peripheral artery disease | I73.9 | 27,980 | 411,754.38 | 56,423.06 |
| **Total outpatient cost attributable to SAH** | - | 725,638 | 14,155,202.77 | 3,226,435.53 |
| **Hospital care** | | | | |
| Coronary artery disease | I25.1 | 6,230 | 23,580,648.17 | 2,978,458.55 |
| Heart failure | I50 | 198,973 | 159,047,613.68 | 20,089,215.61 |
| Subarachnoid hemorrhage | I60 | 10,030 | 29,067,409.02 | 7,991,851.44 |
| Intracerebral hemorrhage | I61 | 13,961 | 21,367,184.35 | 8,468,112.00 |
| Stroke | I64 | 163,076 | 104,906,917.68 | 39,031,302.18 |
| Carotid atherosclerosis | I65.2, I70.8 | 5,586 | 8,112,561.22 | 1,306,549.98 |
| Abdominal aortic aneurysm | I71.3, I71.4 | 3,614 | 20,325,286.16 | 2,749,244.44 |
| Peripheral artery disease | I73.9 | 11,033 | 11,076,037.98 | 1,517,759.10 |
| **Total hospital cost attributable to SAH** | - | 412,503 | 377,483,658.25 | 84,132,493.31 |
| **Total cost attributable to SAH** | - | 1,138,141 | 391,638,861.02 | 87,358,928.84 |

Source: Outpatient Information System (SIA/SUS) and Hospital Information System (SIH/SUS) [43]. Brazil, 2019. Note: the cost attributable to each SAH-related disease was obtained by multiplying the total cost of the disease by its respective PAR value.

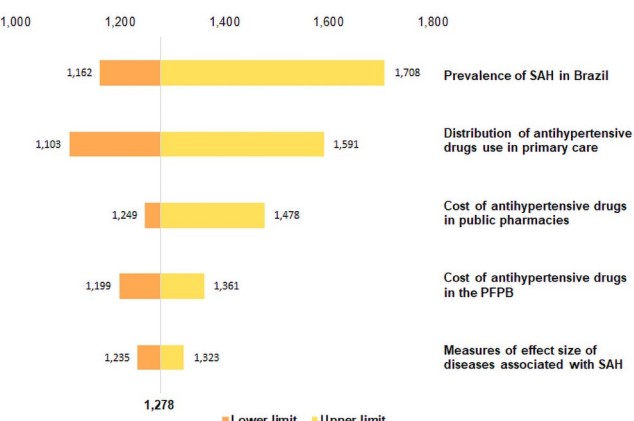

**Fig 2. Tornado diagram of total SAH cost and its complications in the SUS, in millions (Int$).**

lower limit cost of antihypertensives in primary care, and Int$ 776.2 million, the upper limit cost of SAH prevalence in Brazil. Prevalence was the parameter that most varied in sensitivity analysis, representing approximately 36% of the total cost variation of SAH and its complications in the SUS in 2019.

## Discussion

This study estimated that 20,891,905 hypertensive patients were using antihypertensive drugs supplied by the SUS in 2019. The total cost of SAH and the proportion of circulatory system diseases attributable to SAH in Brazil in 2019 was Int$ 581,135,374.73, representing 0.35% of total public spending on health in 2019 [63]. This amount varied between Int$ 501,553,022.21 and Int$ 776,183,338.06 in sensitivity analyses. Assessment of only SAH costs at all healthcare levels (Int$ 493,776,445.89) revealed that 97.3% were in primary care, primarily due to antihypertensive drugs provided free of charge by the SUS (Int$ 363,888,540.14).

One Brazilian study estimated that in 2005 the direct annual cost of SAH was approximately US$ 389.9 million (BRL 947.46 million) for the SUS [25]. However, the study considered a decision tree that simulated the resources used based on Brazilian guidelines for SAH [64] and the opinion of specialists. Moreover, drug costs do not reflect the real values paid for public purchases [25]. On the other hand, our study was based on real world evidence of Brazilian primary care, in terms of the number of medical consultations conducted and the values paid by the SUS. In addition, the present study was based on the usage frequency estimates of antihypertensive drugs available in the SUS obtained in a national survey conducted with 41,433 individuals [51]. Another study that estimated the costs attributable to SAH in Brazil found a value of BRL 2.03 billion (Int$ 923 million) in 2018 [26]. However, it considered more diseases associated with SAH than in our investigation, based on a single study that performed meta-analyses of SAH-related mortality [65]. With respect to primary care, the authors included only the PFPB costs of SAH [26].

A systematic review identified 18 studies that analyzed the economic burden of SAH in LMICs, indicating a median monthly cost of Int$ 22.00 per hypertensive patient [66]. Our study showed an average monthly cost of Int$ 1.97 per hypertensive in the SUS. This difference can be explained by large-scale savings in the SUS, a universal comprehensive public health system, and due to the heterogeneity of the methods and populations of the studies included in the review. Furthermore, while public health spending of a majority of LMICs is proportionally lower [67], Brazil's health system has been predominantly public for over 30 years [68].

Our results are influenced mainly by the cost of primary care, in which patients with low therapeutic complexity predominate. However, our study indicates that the cost per episode of outpatient and hospital care in the SUS is Int\$ 10.49 and Int\$ 166.77, respectively.

In Brazil, the creation of a national SAH registry and the expanded access to essential drugs provided free of charge, including antihypertensives, demonstrated the concern of the SUS with SAH. SAH became the main protagonist in Brazilian primary care with the Family Health Strategy (FHS), considered an example of the global strategy to prevent and manage noncommunicable diseases [69]. The FHS has been associated with a decrease in primary care-related hospitalizations, including SAH, and a decline in mortality throughout the country [70–73]. This concern of addressing SAH in primary care has also been observed in other LMICs, as demonstrated in a scoping review [74]. Healthcare service organization was the most frequent strategy used by the countries analyzed in the review, including organization and equipping healthcare, self-management by education and self-monitoring, and continuity and coordination of actions. Controlled clinical trials have shown that community health interventions involving NPHWs and family physicians were more effective than the usual care provided in LMICs, demonstrating a satisfactory reduction in the blood pressure and cardiovascular risk of hypertensive patients [75–77]. Moreover, community health interventions aimed at SAH are cost saving in both high-income and LMICs [78,79].

In addition to acting in the FHS, providing drugs free of charge is important for managing SAH in Brazilian primary care. Since its expansion to accredited private pharmacies, the PFPB has broadened access to essential drugs, resulting in fewer hospitalizations and death from SAH between 2003 and 2016 [80]. Nevertheless, the Program has been criticized, especially with respect to its costs [34]. One study found higher PFPB drug costs in a Brazilian state capital, with an average difference of 279.8% when compared to the acquisition values of public pharmacies. In relation to drugs for SAH, the difference in costs varied between 119.2% for propranolol and 1,389.4% for captopril [81], and higher in the PFPB. In the present study, these differences ranged between 131.0% for propranolol and 448.4% for enalapril. It is important to underscore that the difference between the two modes of supply may be smaller, since we considered only the drug acquisition values of public pharmacies, excluding logistics and dispensing costs. In a study conducted in Rio de Janeiro state, adding logistics and dispensing costs raised acquisition costs by 70.1%, based on the 25 drugs selected [81].

## Strengths and limitations

Our study systematized real world evidence, which provided a more complete picture of SAH costs in the SUS, including the proportion attributed to circulatory system diseases. In addition, the overview obtained evidence using systematic reviews with meta-analysis to calculate the PAR, avoiding arbitrary selection of the measures of effect size of diseases associated with SAH. However, it is important to emphasize a number of limitations in the present study. First, it was not possible to estimate the costs of preventing and screening SAH in Brazil, since these actions commonly occur during management of other clinical conditions. Second, due to the aggregate funding of primary care in Brazil, we were unable to accurately determine the specific SAH costs in primary care. As such, it was not possible to measure the cost of medical consultations and treatments by NPHWs in primary care, which prompted the use of the value paid for consultations/care in secondary and tertiary care. Third, the logistics costs of public pharmacy drugs were not considered due to the difficulty in measuring them, which may have underestimated these costs. Fourth, the public pharmacy values were used to estimate the costs of drugs not available in the PFPB, which may not be compatible with the real values. Fifth, the costs and proportion of SAH-related circulatory system diseases in secondary and tertiary care

were obtained from administrative databases used to record the clinical procedure and service costs reimbursed by the Ministry of Health of Brazil, which may not reflect the real costs incurred by health services. Sixth, the overview conducted was restricted to systematic reviews with meta-analysis, due to the high number of articles on the topic. However, these were of low quality, and should be interpreted with caution, and no systematic reviews were found for all the complications of SAH described in the literature. Thus, the limitations suggest that the costs of SAH and the complications of the disease in the circulatory system are underestimated in Brazil.

## Implications for public policies

Based on our findings, the major SAH costs were incurred in primary health care (consultations and drugs). Drugs and consultations in primary health care accounted for 62.6% and 20.0% of total SAH costs, respectively. Costs related to complications of SAH were relatively low in our study, probably due to the SAH care coverage in primary health care, provided free of charge. The Brazilian experience in recent years has demonstrated a national effort to manage SAH in primary care, especially due to increased access to essential antihypertensive drugs free of charge and the actions of the FHS. The WHO indicated that in 2019 the availability of essential antihypertensives in middle-income countries such as Brazil was around 92% [82], which is similar to the total access of 97.9% estimated by a national survey [15]. This is also the result of the long trajectory of the PFPB, where a partnership was established with the private sector to improve drug access by hypertensive individuals in Brazil [34]. In addition, the experience of the FHS, an example of the prevention and management of nontransmissible diseases worldwide [69], has demonstrated the importance of the continuous funding of primary care in order to ensure the sustainability of coverage over time [37].

However, a number of challenges to managing SAH in Brazil remain. It will be difficult to reach the goal of a 25% reduction in the prevalence of SAH by 2025 [11], established by the WHO [16], in projections for Brazil [83]. The control of SAH still needs to be improved in many regions of the country [84–87], as well as in other LMICs [19]. Despite efforts to improve the follow-up of hypertensive patients in primary care [88], the weak link between them and primary care professionals [89] and nonattendance at medical consultations [90] are still problems observed in Brazil, compromising the effectiveness of SAH management. As determined in the present study, the number of medical consultations for hypertensive individuals was approximately two-thirds of the estimated number of hypertensive patients indicated for drug treatment in 2019. This represents less than 1 consultation per patient. In addition, a Brazilian survey demonstrated that 10.6% of hypertensive patients intentionally abandoned drug treatment [15]. Experiences in low-income communities in Brazil have shown opportunities for improving SAH management in primary care, such as investing in the training of health professionals and greater multidisciplinary engagement. Barriers include health system restrictions and local dietary culture [91].

Given the restricted budget for public health in the last 20 years [92], national and international decision makers could use the results of this study to formulate public policies and optimize resources related to SAH. Moreover, cost-of-illness studies like ours can complement economic evaluations conducted in health technology assessment, such as cost-effectiveness and budget impact analysis.

Finally, it is important to note that there is still a need to improve the health information systems used to manage diseases in Brazil [27]. Disparities remain in the health information records of the different Brazilian institutions, in addition to difficulties related to the technological infrastructure and qualification of the professionals involved in data collection [93].

Thus, it is important to continuously modernize Brazilian information systems, as well as professional qualification and awareness of the relevance of ensuring that the health registers are complete.

## Supporting information

**S1 Fig. Flowchart of the overview conducted.**
(TIF)

**S1 Table. Estimated hypertensive population using antihypertensive drugs in SUS primary care.** Brazil, 2019.
(DOCX)

**S2 Table. Estimated costs of antihypertensive drugs from public pharmacies in primary care.** Brazil, 2019.
(DOCX)

**S3 Table. Estimated costs of antihypertensive drugs from the Brazilian Popular Pharmacy Program (PFPB) in primary care.** Brazil, 2019.
(DOCX)

**S4 Table. Distribution of use and estimated total cost of antihypertensive drugs in primary care.** Brazil, 2019.
(DOCX)

**S5 Table. Search strategies in scientific databases.**
(DOCX)

**S6 Table. Critical appraisal of the systematic review studies using AMSTAR 2.**
(DOCX)

## Author Contributions

**Conceptualization:** Daniel da Silva Pereira Curado, Everton Nunes da Silva.

**Formal analysis:** Daniel da Silva Pereira Curado, Dalila Fernandes Gomes, Thales Brendon Castano Silva, Paulo Henrique Ribeiro Fernandes Almeida, Noemia Urruth Leão Tavares, Everton Nunes da Silva.

**Investigation:** Daniel da Silva Pereira Curado, Dalila Fernandes Gomes, Thales Brendon Castano Silva, Paulo Henrique Ribeiro Fernandes Almeida.

**Methodology:** Daniel da Silva Pereira Curado, Everton Nunes da Silva.

**Supervision:** Everton Nunes da Silva.

**Validation:** Daniel da Silva Pereira Curado, Noemia Urruth Leão Tavares, Everton Nunes da Silva.

**Visualization:** Daniel da Silva Pereira Curado, Everton Nunes da Silva.

**Writing – original draft:** Daniel da Silva Pereira Curado.

**Writing – review & editing:** Daniel da Silva Pereira Curado, Dalila Fernandes Gomes, Thales Brendon Castano Silva, Paulo Henrique Ribeiro Fernandes Almeida, Noemia Urruth Leão Tavares, Camila Alves Areda, Everton Nunes da Silva.

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
