## [Decision Letter · Decision Letter 0]

24 Feb 2021

PONE-D-20-37639

Cost of systemic arterial hypertension and its complications in the circulatory system from the perspective of the Brazilian public health system in 2019

PLOS ONE

Dear Dr. Curado,

Thank you for submitting your manuscript to PLOS ONE. After careful consideration, we feel that it has merit but does not fully meet PLOS ONE’s publication criteria as it currently stands. Therefore, we invite you to submit a revised version of the manuscript that addresses the points raised during the review process.

We look forward to receiving your revised manuscript.

Kind regards,

Rosa Maria Urbanos Garrido, PhD

Academic Editor

PLOS ONE

Reviewers' comments:

**Comments to the Author**

1. Is the manuscript technically sound, and do the data support the conclusions?

Reviewer #1: Yes

Reviewer #2: Partly

2. Has the statistical analysis been performed appropriately and rigorously? 

Reviewer #1: Yes

Reviewer #2: No

3. Have the authors made all data underlying the findings in their manuscript fully available?

Reviewer #1: Yes

Reviewer #2: Yes

4. Is the manuscript presented in an intelligible fashion and written in standard English?

Reviewer #1: No

Reviewer #2: Yes

5. Review Comments to the Author

Reviewer #1: Cost of systemic arterial hypertension and its complications in the circulatory system from the perspective of the Brazilian public health system in 2019

The study estimates the cost of Systemic Arterial Hypertension (SAH) and circulatory system diseases attributable to SAH from the perspective of the Brazilian public health system in 2019. The authors developed a prevalence-based cost-of-illness using a top-down approach.

General comments:

1) Section “Study setting” (line 112 and 121)

The authors have stated that currently, 75% of Brazilians depend exclusively on the public system namely SUS (the source of your). The SAH prevalence presented is from the entire Brazilian population. Therefore, it seems that your PAF’s were calculated for the entire population. Is the prevalence the same between populations relying on public and private healthcare systems in Brazil? Is the survey considering the private and public health systems’ users?

2) Section “Study design” (line 124)

It seems that the reference 15 needs to be corrected. The reference 15 is not a national survey published in 2019.

3) Section “Costs of SAH-related circulatory system diseases” (line 174)

The section’s order is a bit confusing. First, the authors stated that they used quantitative evidence of effect size from systematic reviews, secondly, they explained the systematic review development. I’m guessing the authors needed to first perform the systematic review and then, with the results, find which diseases with SAH as a risk factor they have the quantitative evidence. The way it is, it looks like they are talking about different systematic reviews.

4) Section “ethical aspects” (line 238)

The text could be briefer, for instance “this study used only secondary data available in public databases, therefore approval from the Research Ethics Committee of the National Research Ethics Commission (CEP/CONEP) was not required.

5) English review: the paper is not free of grammar and punctuation mistakes. Authors may want to consider consulting a proof-reading service. (some suggestions below)

Line 50: “in developed and developing countries”. Sometimes the authors used low- and middle-income countries (LMIC’s). It is better if they use only one term (preferably LMIC)

line 354: By economies of scale of the SUS, do you mean large-scale savings? Please, make the sentence more clear

line 370: Healthcare service organization was the most frequently used strategy by the countries – replace with […] was the most frequently strategy used by the countries

line 433: use “remains” instead of “remain”

line 436: replace “parts of the country” with “country regions”

Thank you for the opportunity to review this interesting and important work!

Reviewer #2: Authors,

Congratulations for your analysis on costs of Hypertension in Brazil. In order to improve your presentation, I have some comments to be considered:

1. The topic of direct costs should be included in the tittle.

2. In the abstract we need an idea of the impact of this illness in Brazil.

3. It should be clear that major costs are related to complications. An effective primary care management could reduce these costs.

4. I find optimistic your description of the SUS. Some evaluations show limitations in drugs availability, difficulties to allocate human resources, and also barriers to health care facilities. How did your evaluation considered this point?

5. Your estimations on drugs prescribed and used could be undeestimated. Karnikowski, M., Nóbrega, O., Naves, J. et al. Access to Essential Drugs in 11 Brazilian Cities: A Community-based Evaluation and Action Method. J Public Health Pol 25, 288–298 (2004). https://doi.org/10.1057/palgrave.jphp.3190029

6. How can we know about barriers to health care without considering or estimating out-of-pocket expenditure of patients and their families?

7. It is not clear how complications were estimated, and why kidney failure was not considered.

8. Without comparison parametres, it is difficult for me to have a clear idea about the costs. If you want a wide audience please help us to have an idea comparing, showing tendences, etc. As it is written looks for Brazilian readers.

9. If there are some other studies, why don't you replicate some methods to make these analysis comparable.

10. How did you standarized regional or individual differences?

11. Without a national catalogue, How can we know what drugs were used in how many patients?

12. It should be specified if literature comparisons were done for brazilian cases or worlwide? How do you deal with econpomical and political differences, in terms of health system response?

With kind regards,

6. PLOS authors have the option to publish the peer review history of their article (what does this mean?). If published, this will include your full peer review and any attached files.

Reviewer #1: No

Reviewer #2: **Yes: **Emanuel Orozco-Nuñez, Ma.

---

## [Author Response · Author response to Decision Letter 0]

15 Apr 2021

Editor, comment 1: “Please ensure that your manuscript meets PLOS ONE's style requirements, including those for file naming. The PLOS ONE style templates can be found at https://journals.plos.org/plosone/s/file?id=wjVg/PLOSOne_formatting_sample_main_body.pdf and https://journals.plos.org/plosone/s/file?id=ba62/PLOSOne_formatting_sample_title_authors_affiliations.pdf “.

Our response: Thank you for the information.

Editor, comment 2: “Please include captions for your Supporting Information files at the end of your manuscript, and update any in-text citations to match accordingly. Please see our Supporting Information guidelines for more information: http://journals.plos.org/plosone/s/supporting-information.”

Our response: Thank you. We followed the PLOS ONE Supporting Information guidelines.

Reviewer 1, comment 1: “Cost of systemic arterial hypertension and its complications in the circulatory system from the perspective of the Brazilian public health system in 2019. The study estimates the cost of Systemic Arterial Hypertension (SAH) and circulatory system diseases attributable to SAH from the perspective of the Brazilian public health system in 2019. The authors developed a prevalence-based cost-of-illness using a top-down approach.”

Our Response: We would like to thank you for your comments and the time spent reviewing our manuscript.

Reviewer 1, comment 2: “General comments. Section “Study setting” (line 112 and 121). The authors have stated that currently, 75% of Brazilians depend exclusively on the public system namely SUS (the source of your). The SAH prevalence presented is from the entire Brazilian population. Therefore, it seems that your PAF’s were calculated for the entire population. Is the prevalence the same between populations relying on public and private healthcare systems in Brazil? Is the survey considering the private and public health systems’ users?”

Our Response: Thank you for your comment. We used the SAH prevalence for the entire Brazilian population to calculate the Population Attributable Risk (PAR). We did not find any population study that estimated SAH prevalence stratified by private and public health system users. However, when we calculated the attributable costs of the circulatory system diseases (PAR*total cost of disease), we used only the total costs of public health system users; the same applies to the costs of drugs used at the primary healthcare level. 

We found one study based on a telephone survey [1], but it is only representative of the 26 state capitals and the Federal District. Moreover, the prevalence of SAH in public and private health system users was similar but significant (25.3 (95% CI 24.2−26.3) versus 22.8 (95% CI 21.9−23.7), respectively). It did not bias our results because our sensitivity analyses cover this range of SAH prevalence. 

[1] Malta DC, Bernal RTI, Andrade SSCA, Silva MMAD, Velasquez-Melendez G. Prevalence of and factors associated with self-reported high blood pressure in Brazilian adults. Rev Saude Publica. 2017 Jun 1;51(suppl 1):11s. DOI: 10.1590/S1518-8787.2017051000006. PMID: 28591346; PMCID: PMC5676350.

Reviewer 1, comment 3: “Section “Study design” (line 124).

It seems that the reference 15 needs to be corrected. The reference 15 is not a national survey published in 2019.”

Our Response: Thank you for your comment. In cost-of-illness studies, it is important to report the year for which the costs were adjusted for inflation. To make it clear, we changed the sentence to:

“Moreover, only the adult population was considered (≥20 years of age) in analysis (39), which was obtained from the main national survey (15). This study was conducted from the perspective of the SUS and costs were adjusted to 2019 prices.”

Reviewer 1, comment 4: “Section “Costs of SAH-related circulatory system diseases” (line 174). The section’s order is a bit confusing. First, the authors stated that they used quantitative evidence of effect size from systematic reviews, secondly, they explained the systematic review development. I’m guessing the authors needed to first perform the systematic review and then, with the results, find which diseases with SAH as a risk factor they have the quantitative evidence. The way it is, it looks like they are talking about different systematic reviews.”

Our Response: Thank you for your comment. We changed the order of the reporting, according to your suggestion (we reported the diseases included after explaining the overview of systematic review development).

Reviewer 1, comment 5: “Section “ethical aspects” (line 238).

The text could be briefer, for instance “this study used only secondary data available in public databases, therefore approval from the Research Ethics Committee of the National Research Ethics Commission (CEP/CONEP) was not required.”

Our Response: Thank you for your comment. We changed the text as suggested by the reviewer. 

Reviewer 1, comment 5: “English review: the paper is not free of grammar and punctuation mistakes. Authors may want to consider consulting a proof-reading service. (some suggestions below). Line 50: “in developed and developing countries”. Sometimes the authors used low- and middle-income countries (LMIC’s). It is better if they use only one term (preferably LMIC); line 354: By economies of scale of the SUS, do you mean large-scale savings? Please, make the sentence more clear; line 370: Healthcare service organization was the most frequently used strategy by the countries – replace with […] was the most frequently strategy used by the countries; line 433: use “remains” instead of “remain”; line 436: replace “parts of the country” with “country regions”.”

Our response: Thank you for your comment. As suggested by the reviewer, we used an academic proofreading service with an English native speaker and have provided a certificate attesting to this fact. 

Reviewer 1, comment 6: “Thank you for the opportunity to review this interesting and important work!”

Our response: Thank you again for your helpful comments. 

Reviewer 2, comment 1: “Authors, Congratulations for your analysis on costs of Hypertension in Brazil. In order to improve your presentation, I have some comments to be considered:”

Our response: We would like to thank you for your comments and the time spent reviewing our manuscript.

Reviewer 2, comment 2: “The topic of direct costs should be included in the tittle.”

Our response: Thank you for your comment. As suggested by the reviewer, we included “direct costs” in the title, as shown below:

“Direct cost of systemic arterial hypertension and its complications in the circulatory system from the perspective of the Brazilian public health system in 2019”

Reviewer 2, comment 3: “In the abstract we need an idea of the impact of this illness in Brazil.”

Our response: Thank you for your comment. As suggested by the reviewer, we added information about Brazil. 

“Introduction: Systemic arterial hypertension (SAH), a global public health problem and the primary risk factor for cardiovascular diseases, has a significant financial impact on health systems. In Brazil, the prevalence of SAH is 23.7%, which caused 203,000 deaths and 3.9 million DALYs in 2015.”

Reviewer 2, comment 4: “It should be clear that major costs are related to complications. An effective primary care management could reduce these costs.”

Our response: Thank you for your comment. Based on our findings, the major SAH costs were incurred in primary health care (consultations and drugs). Drugs and consultations in primary health care accounted for 62.6% and 20.0% of total SAH costs, respectively. Costs related to complications of SAH were relatively low in our study, probably due to the substantial SAH coverage in primary health care, provided free of charge. We included this information in the Discussion in “Implications for public policies.”

Reviewer 2, comment 5: “I find optimistic your description of the SUS. Some evaluations show limitations in drugs availability, difficulties to allocate human resources, and also barriers to healthcare facilities. How did your evaluation considered this point?”

Our response: Thank you for your comment. With a view to portraying the SUS as accurately as possible, we used real-world evidence, such as data from the SUS information systems (SIA/SUS, SIH/SUS and SISAB/SUS), and PNAUM, the national survey. Moreover, we pointed out the recent progress in Brazil in the management of SAH in primary care. However, as in other public health systems, the SUS is not without problems involving access to care. Thus, in our discussion we describe the limitations of our analysis and the opportunities of improving the SUS.

Reviewer 2, comment 6: “Your estimations on drugs prescribed and used could be undeestimated. Karnikowski, M., Nóbrega, O., Naves, J. et al. Access to Essential Drugs in 11 Brazilian Cities: A Community-based Evaluation and Action Method. J Public Health Pol 25, 288–298 (2004). https://doi.org/10.1057/palgrave.jphp.3190029”

Our response: Thank you for your comment. In order to estimate the use of each antihypertensive drug, we used the PNAUM, a national population-based cross-sectional study that assessed access and the use of drugs by the Brazilian population, including hypertensive patients. This survey was performed in urban households in the five regions of the country between September 2013 and February 2014, nearly 10 years after the study by Karnikowski et al. (2004). We also conducted sensitivity analyses where we varied the distribution of antihypertensive medication in primary care based on the 95% confidence intervals of the PNAUM (Table S4).

Reviewer 2, comment 7: “How can we know about barriers to health care without considering or estimating out-of-pocket expenditure of patients and their families?”

Our response: Thank you for your comment. Although we recognize the importance of identifying access barriers to health care, it is beyond the scope of our study. Our objective was to provide a real-world estimate of SAH costs in the public health system. To investigate barriers to health care, we would need another methodological approach, such as a qualitative study or qualitative systematic review. 

Reviewer 2, comment 8: “It is not clear how complications were estimated, and why kidney failure was not considered.”

Our response: Thank you for your comment. We conducted an overview of systematic reviews to identify those that reported measures of effect size (relative risk, odds ratio or hazard ratio) of individuals with SAH who developed any disease versus those without SAH. This information is required to calculate the population attributable risk (PAR). Unfortunately, we did not find any systematic review that provided this information related to kidney failure. We justified our decision to select only systematic review as a limitation of our study, as shown below: 

“Sixth, the overview conducted was restricted to systematic reviews with meta-analysis, due to the high number of articles on the topic. However, these were of low quality, and should be interpreted with caution, and no systematic reviews were found for all the complications of SAH described in the literature. Thus, the limitations suggest that the costs of SAH and the complications of the disease in the circulatory system are underestimated in Brazil.”

Reviewer 2, comment 9: “Without comparison parametres, it is difficult for me to have a clear idea about the costs. If you want a wide audience please help us to have an idea comparing, showing tendences, etc. As it is written looks for Brazilian readers.”

Our response: Thank you for your comment. The dissemination plan of our study includes national and international readers. As such, we chose PlOS ONE, a high-quality journal with open access. We respectfully disagree with the reviewer, since we have provided several parameters that can be compared with other studies conducted worldwide. First, we provided an estimate of the economic burden of SAH in terms of public spending on health (0.35%). This parameter indicates the impact of SAH on the public health budget. However, we did not find any study conducted worldwide that calculated this parameter. Second, we reported cost estimates stratified by type of cost, such as primary outpatient and inpatient health care (consultations and drugs). The proportion of each cost can also be compared with other studies worldwide. Third, we quoted several cost-of-illness studies carried out worldwide, but they did not provide the parameters we calculated based on Brazilian data. Finally, cost-of-illness studies varied considerably in terms of methodological approach, data availability, and healthcare system funding, which pose several challenges in terms of comparing the results between studies. 

Reviewer 2, comment 10: “If there are some other studies, why don't you replicate some methods to make these analysis comparable.”

Our response: Thank you for your comment. We followed national and international methodological guidelines on cost-of-illness studies. Guidelines are better benchmarks than a single study. As we highlighted in comment 9, cost-of-illness studies varied considerably in terms of methodological approach, data availability, and healthcare system funding. On this basis, we are confident that we applied the best practices available in the literature. Moreover, our estimate reflects the data publicly available in Brazil. 

Reviewer 2, comment 11: “How did you standarized regional or individual differences?

Our response: Thank you for your comment. We used data from the national SUS information systems (SIA/SUS, SIH/SUS and SISAB/SUS), and the PNAUM survey, a national population-based cross-sectional study carried out in Brazil. Since, the aggregate data represent the entire Brazilian population, we did not need to standardize regional or individual differences.

Reviewer 2, comment 12: “Without a national catalogue, How can we know what drugs were used in how many patients?”

Our response: Thank you for your comment. There is a national drug catalogue in the public healthcare system (SUS). We included this information in the Methods section.

“In addition, the SUS provides essential medication based on a national drug catalog (including antihypertensives, such as hydrochlorothiazide, losartan, captopril, enalapril, atenolol, amlodipine, propranolol, furosemide and nifedipine) (30) free of charge at public and private pharmacies accredited in the Brazilian Popular Pharmacy Program (PFPB) (31).”

With respect to the use of each drug by population, we obtained this information from the PNAUM population study.

Reviewer 2, comment 13: “It should be specified if literature comparisons were done for brazilian cases or worlwide? How do you deal with econpomical and political differences, in terms of health system response?”

Our response: Thank you for your comment. Please see our response in “Reviewer 2, comment 9”.

---

## [Decision Letter · Decision Letter 1]

19 May 2021

PONE-D-20-37639R1

Direct cost of systemic arterial hypertension and its complications in the circulatory system from the perspective of the Brazilian public health system in 2019

PLOS ONE

Dear Dr. Curado,

Thank you for submitting your manuscript to PLOS ONE. After careful consideration, we feel that it has merit but does not fully meet PLOS ONE’s publication criteria as it currently stands. Therefore, we invite you to submit a revised version of the manuscript that addresses the points raised during the review process.

We look forward to receiving your revised manuscript.

Kind regards,

Rosa Maria Urbanos Garrido, PhD

Academic Editor

PLOS ONE

Journal Requirements:

Reviewers' comments:

Reviewer's Responses to Questions

**Comments to the Author**

1. If the authors have adequately addressed your comments raised in a previous round of review and you feel that this manuscript is now acceptable for publication, you may indicate that here to bypass the “Comments to the Author” section, enter your conflict of interest statement in the “Confidential to Editor” section, and submit your "Accept" recommendation.

Reviewer #2: All comments have been addressed

2. Is the manuscript technically sound, and do the data support the conclusions?

Reviewer #2: Yes

3. Has the statistical analysis been performed appropriately and rigorously? 

Reviewer #2: Yes

4. Have the authors made all data underlying the findings in their manuscript fully available?

Reviewer #2: Yes

5. Is the manuscript presented in an intelligible fashion and written in standard English?

Reviewer #2: Yes

6. Review Comments to the Author

Reviewer #2: Dear Authors,

Congratulations for this important scientific communication. I appreciate your attention to my previous requests. Nevertheless, two minor observations prevail: 1) on page 5, line 99, you state that the SUS provides essential mecidations, when there are evaluations based on national data that show limitations in coverage of health care (https://www.scielosp.org/article/rsp/2017.v51suppl1/3s/#). 2) On page 21, line 423, you suggest that coverage of primary care is succesful for SAH. Again, some sources show that mortality due to this health need is considerable (https://www.scielosp.org/article/rsp/2014.v48n4/671-681/en/#). Perhaps you could suggest a way to achieve a better scenario.

With kind regards,

7. PLOS authors have the option to publish the peer review history of their article (what does this mean?). If published, this will include your full peer review and any attached files.

Reviewer #2: **Yes: **Emanuel Orozco, Ma.

---

## [Author Response · Author response to Decision Letter 1]

25 May 2021

Editor, comment 1: “Please review your reference list to ensure that it is complete and correct. If you have cited papers that have been retracted, please include the rationale for doing so in the manuscript text, or remove these references and replace them with relevant current references. Any changes to the reference list should be mentioned in the rebuttal letter that accompanies your revised manuscript. If you need to cite a retracted article, indicate the article’s retracted status in the References list and also include a citation and full reference for the retraction notice. “

Our response: Thank you for the information. Our reference list is complete and correct.

Reviewer 2, comment 1: “Dear Authors, congratulations for this important scientific communication. I appreciate your attention to my previous requests. Nevertheless, two minor observations prevail: 1) on page 5, line 99, you state that the SUS provides essential medications, when there are evaluations based on national data that show limitations in coverage of health care (https://www.scielosp.org/article/rsp/2017.v51suppl1/3s/#). 2) On page 21, line 423, you suggest that coverage of primary care is successful for SAH. Again, some sources show that mortality due to this health need is considerable (https://www.scielosp.org/article/rsp/2014.v48n4/671-681/en/#). Perhaps you could suggest a way to achieve a better scenario.”

Our response: Thank you for the comments. We made the suggested changes. 1) On page 5, line 105, we added the information about barriers to access to medication in Brazil, as follows:

“Although there is a legal obligation to provide drugs free of charge, the population has faced barriers to access to medication in Brazil, not achieving full coverage (35).”

2) We moderated the statement on the contribution of primary care to SAH management in Brazil, as the reviewer suggested. We removed the word “substantial” from the following sentence: 

"Costs related to complications of SAH were relatively low in our study, probably due to the SAH care coverage in primary health care, provided free of charge.” 

In regard to the article cited by the reviewer, we added information on improving health information systems to page 23, line 462 of the revised manuscript:

“Finally, it is important to note that there is still a need to improve the health information systems used to manage diseases in Brazil (27). Disparities remain in the health information records of the different Brazilian institutions, in addition to difficulties related to the technological infrastructure and qualification of the professionals involved in data collection (93). Thus, it is important to continuously modernize Brazilian information systems, as well as professional qualification and awareness of the relevance of ensuring that the health registers are complete.”

---

## [Editor Report · Decision Letter 2]

28 May 2021

Direct cost of systemic arterial hypertension and its complications in the circulatory system from the perspective of the Brazilian public health system in 2019

PONE-D-20-37639R2

Dear Dr. Curado,

We’re pleased to inform you that your manuscript has been judged scientifically suitable for publication and will be formally accepted for publication once it meets all outstanding technical requirements.

Kind regards,

Rosa Maria Urbanos Garrido, PhD

Academic Editor

PLOS ONE
---

## [Editor Report · Acceptance letter]

2 Jun 2021

PONE-D-20-37639R2 

Direct cost of systemic arterial hypertension and its complications in the circulatory system from the perspective of the Brazilian public health system in 2019 

Dear Dr. Curado:

I'm pleased to inform you that your manuscript has been deemed suitable for publication in PLOS ONE. Congratulations! Your manuscript is now with our production department. 

Kind regards, 

on behalf of

Dr. Rosa Maria Urbanos Garrido 

Academic Editor

PLOS ONE